# Path integration in large-scale space and with novel geometries: Comparing vector addition and encoding-error models

Sevan K. Harootonian[1,2¤], Robert C. Wilson[2,3,4], Lukáš Hejtmánek[1,5], Eli M. Ziskin[1,2], Arne D. Ekstrom[1,2,4]*

1 Center for Neuroscience, University of California Davis, Davis, California, United States of America, 2 Psychology Department, University of Arizona, Tucson, Arizona, United States of America, 3 Cognitive Science Program, University of Arizona, Tucson, Arizona, United States of America, 4 Evelyn McKnight Brain Institute, University of Arizona, Tucson, Arizona, United States of America, 5 Third Faculty of Medicine, Charles University, Ruská, Prague, Czech Republic

¤ Current address: Psychology Department, Princeton University, Princeton, New Jersey, United States of America

* adekstrom@email.arizona.edu

**Data Availability Statement:** All data files are available at: github.com/sharootonian/PA-TCT.

**Funding:** Research supported by grants from NSF Division of Behavioral and Cognitive Sciences

## Abstract

Path integration is thought to rely on vestibular and proprioceptive cues yet most studies in humans involve primarily visual input, providing limited insight into their respective contributions. We developed a paradigm involving walking in an omnidirectional treadmill in which participants were guided on two sides of a triangle and then found their back way to origin. In Experiment 1, we tested a range of different triangle types while keeping the distance of the unguided side constant to determine the influence of spatial geometry. Participants overshot the angle they needed to turn and undershot the distance they needed to walk, with no consistent effect of triangle type. In Experiment 2, we manipulated distance while keeping angle constant to determine how path integration operated over both shorter and longer distances. Participants underestimated the distance they needed to walk to the origin, with error increasing as a function of the walked distance. To attempt to account for our findings, we developed configural-based computational models involving vector addition, the second of which included terms for the influence of past trials on the current one. We compared against a previously developed configural model of human path integration, the Encoding-Error model. We found that the vector addition models captured the tendency of participants to under-encode guided sides of the triangles and an influence of past trials on current trials. Together, our findings expand our understanding of body-based contributions to human path integration, further suggesting the value of vector addition models in understanding these important components of human navigation.

## Author summary

How do we remember where we have been? One important mechanism for doing so is called path integration, which refers to the computation of one's position in space with only self-motion cues. By tracking the direction and distance we have walked, we

[BCS-1630296] awarded to Arne Ekstrom. The funders had no role in study design, data collection and analysis, decision to publish, or preparation of the manuscript.

**Competing interests:** The authors have declared that no competing interests exist.

create a mental arrow from the current location to the origin, termed the homing vector. Previous studies have shown that the homing vector is subject to systematic distortions depending on previously experienced paths, yet what influences these patterns of errors, particularly in humans, remains uncertain. In this study, we compare two models of path integration based on participants walking two sides of a triangle without vision and then completing the third side based on their estimate of the homing vector. We found no effect of triangle shape on systematic errors, while the systematic errors scaled with path length logarithmically, similar to Weber-Fechner law. While we show that both models captured participants' behavior, a model based on vector addition best captured the patterns of error in the homing vector. Our study therefore has important implications for how humans track their location, suggesting that vector-based models provide a reasonable and simple explanation for how we do so.

## Introduction

"Dead reckoning," first coined by Charles Darwin [1], described a process whereby experienced navigators kept course to a particular spot over long distances and changes in directions, despite being in the featureless arctic tundra. All animal species tested, including spiders [2], bees [3], gerbils [4], hamsters [5], house mice [6], rats [7], birds [8], ants [9], arthropods [10], drosophila [11], dogs [12], cats [13], and humans [14] show the ability to dead reckon, thought to involve a computational process called path integration (please see [15–17] for a review of prior literature). Because humans employ vision as a primary modality to navigate, research on path integration has often focused on experiments in which visual input provides sufficient information to solve most navigational tasks, such as in desktop virtual reality. This is because visually rendered optic flow provides a velocity signal sufficient for some forms of path integration, with desktop virtual reality providing the opportunity to decouple visual from body-based cues in freely navigating humans. A limitation, however, with desktop VR is that it lacks the enriched cues and codes that we obtain when we freely move our body throughout space, thought to contribute critically to path integration [18–20], and does not allow comparison of the effects of visual vs. body-based inputs on path integration.

Past experiments on path integration have often involved a path completion task in which the navigator is guided in physical space and must return using the shortest trajectory back to the origin with no guidance [2, 21, 22]. Behavioral results both in vertebrates and invertebrates show a systematic bias in the return trajectory which is independent of random accumulated noise [23–27]. Two sets of computational models have been proposed as potential strategies participants might use, in some form, for path integration. The Homing Vector Model (also called a continuous strategy) assumes that the navigator does not encode information about the outbound path but rather continuously updates their current position relative to the origin by computing a continuous homing vector. To capture systematic errors in return paths, Homing Vector Models assume different variations of memory decay or leaky integration which are in theory independent of outbound path configurations [28–30].

Configural Models, in contrast, suggest a continuous encoding of the outbound paths (guided path) which are then manipulated using trigonometric or vector calculations to compute a return trajectory (configural homing vector) when required. One of these models, the Encoding-Error Model, suggests that systematic errors in the return path are due to errors in encoding of the outbound paths and their relative configurations [31, 32]. Recent work, however, suggests that execution errors (during the unguided path) make the largest contribution to systematic errors [33]. Furthermore, Wiener et. al. systematically compared configural and

continuous updating strategies by explicitly telling participants to focus on the outbound path or the origin. The results suggest that participants can deploy either configural or continual strategies during a path completion task [34]. Given the widespread application of configural models to both human and non-human path integration [23, 24, 26, 31, 32, 35] and the evidence that humans employ both configural and homing strategies [34], we directly compared three different configural models of path integration to better understand the patterns of errors that accumulate as participants dead-reckon.

In humans, a frequently employed experimental paradigm is the triangle completion task in which the experimenter guides the participant on two sides of a triangle and then the participant must return, without guidance, to the origin [21, 35]. To model how systematic errors accumulate when human participants perform path completion tasks and specifically the triangle completion task, Fujita et al. 1993 [31] proposed the Encoding-Error Model. This configural model of path integration uses the law of cosines to compute the third side of the triangle given the distances and angles encoded during first two sides (which are the guided portions). Therefore, it assumes that all systematic errors accumulate during encoding. Additionally, the model has four assumptions: (1) the internal representation satisfies Euclidean axioms (2) straight-line segments are encoded as a single value that represents their length (3) turns are encoded as a single value that represents the angle (4) there are no systematic errors during computation or execution of the homeward trajectory.

In support of their model, Fujita et al. fit data collected in [35] and [21] involving the triangle completion task in the absence of vision. The model captured the systematic errors seen in both studies to a relatively high degree (see [31] Table 3). As predicted, though, the model performed poorly for paths with more than two sides or paths that crossed each other. The Encoding-Error model was expanded by Klatzky et al. 1999 [32] to test its generalizability, who found that systematic errors were context and experience dependent. They also found that while partial vision increased path accuracy, it did not change the pattern of errors. Another important finding, supported by the Encoding-Error Model and other studies [36], was that systematic errors in path integration, at least in small environments ($\leq$10m), showed a pattern of regression to the mean. Specifically, past paths influenced the current paths and therefore, shorter angles and distances were overestimated and longer angles and distances were underestimated [21, 35]. Petzschner and Glasauer 2011 [36] (using desktop virtual reality) extended these findings by showing that the same angle or distance could be overestimated in some cases and underestimated in others. The degree of under/overestimation depended on the distribution of priors, known as range effects, such that a broader distribution of priors (e.g., distances from 5–100 meters vs. 5–10 meters) increased the effect of the regression to the mean [37].

The issue of how the distribution of priors influences the current trajectories, however, begs the question of how path configurations affect errors in the triangle completion task. Specifically, past work suggests that the geometric properties of shapes can influence navigation [38, 39]. For example, shapes like isosceles or equilateral triangles could serve as "templates" for how we learn paths [12] by providing a means for estimating paths that approximate it. Grid cells, neurons that fire as animals explore spatial environments, show 6-fold symmetry, with equilateral triangles composing part of this structure [40]. Given arguments that neural codes might manifest in spatial representations useful for navigating [41, 42] and the proposed link between path integration and grid cells [43, 44], it could be the case that geometric regularities (such as equilateral triangles) also influence path integration. Indeed, some past studies on the triangle completion task provide support for the idea that geometric regularities can, in some cases, influence path accuracy [35]. Yet, whether and how different types of triangles (equilateral vs. isosceles vs. scalene) influence path accuracy and patterns of errors on the unguided side in the triangle completion task remains unclear.

Another important yet largely unanswered question about human path integration regards the accuracy and patterns of errors over longer distances. The vast majority of studies in human path integration have involved small-scale environments (< = 10 meters) and consistent with this, computational models of path integration largely base their predictions on much smaller scales. For example, Klatzky et al. 1999 [32] suggested that it was unlikely that the same encoding function in their model was used for pathways that were larger than 10 meters [32]. A more recent computational model of path integration that employs grid cells suggests that, in the absence of specific mnemonic aids, path integration codes may rapidly degrade in mammals [45], consistent with the idea that path integration could breakdown dramatically over longer distances. Interestingly, however, other grid cell models assuming continuous rather than configural encoding suggest reliable estimations up to 100 meters [46]. Thus, an important question to test is how well human participants perform at path integration over longer distances (> = 100 meters).

In the current study, we employed an omnidirectional treadmill and somatosensory input via handheld controllers (Fig 1A) to determine the extent to which manipulating the angle and distance participants needed to walk affected the accuracy of navigation without vision. The unique advantage of using the omnidirectional treadmill is that it permits manipulation of infinitely large spaces thereby eliminating the need for any boundaries while preserving the input from walking. The issue of boundaries, perceived or imagined, is of potential importance because if a participant were to overshoot a distance they would be stopped before hitting a wall, providing inadvertent feedback on the distances of the room and potentially affecting subsequent performance. In addition, the use of handheld controllers allowed us to carefully manipulate participant trajectories on the guided sides, an issue we return to in greater depth in the discussion.

To attempt to capture the pattern of errors and history effects in the triangle completion task in human participants, we created a simple yet novel configural model based on vector addition often used to understand path integration in other species [23, 24]. The model assumes the outbound path is encoded as discrete sets of vectors, such that individual vectors can be employed to compute new trajectories using a configural homing vector. This model allows for a smaller set of required assumptions and free parameters when compared to Encoding-Error Model as well as a physiologically plausible mechanism by means of head direction accumulation [26]. In Experiment 1, we set out to test the extent to which different shapes of triangles (isosceles, equilateral) influenced how participants learned the configural homing vector by explicitly manipulating triangle type (while keeping configural homing distance constant). In Experiment 2, we explicitly manipulated the distance participants had to walk to reach the origin (while keeping triangle type constant) to determine how participants performed over a range of different distances. Critically, by manipulating these variables, we were able to simultaneously test hypotheses related to 1) triangle type and whether some might perform better than others; 2) configural homing distance and whether path integration would show different properties at ~10m vs. ~100m; 3) which model, one based on vector addition or the Encoding-Error model, would provide a better account of the pattern of findings. We provide a detailed comparison of the assumptions and setup of the different models in the Methods section.

## Results

### Experiment 1: *Basic behavior*

An example raw trace of a participant's path overlaid on the vector distances is shown in Fig 1D (dashed lines) between the points. We defined angle error as the difference between the

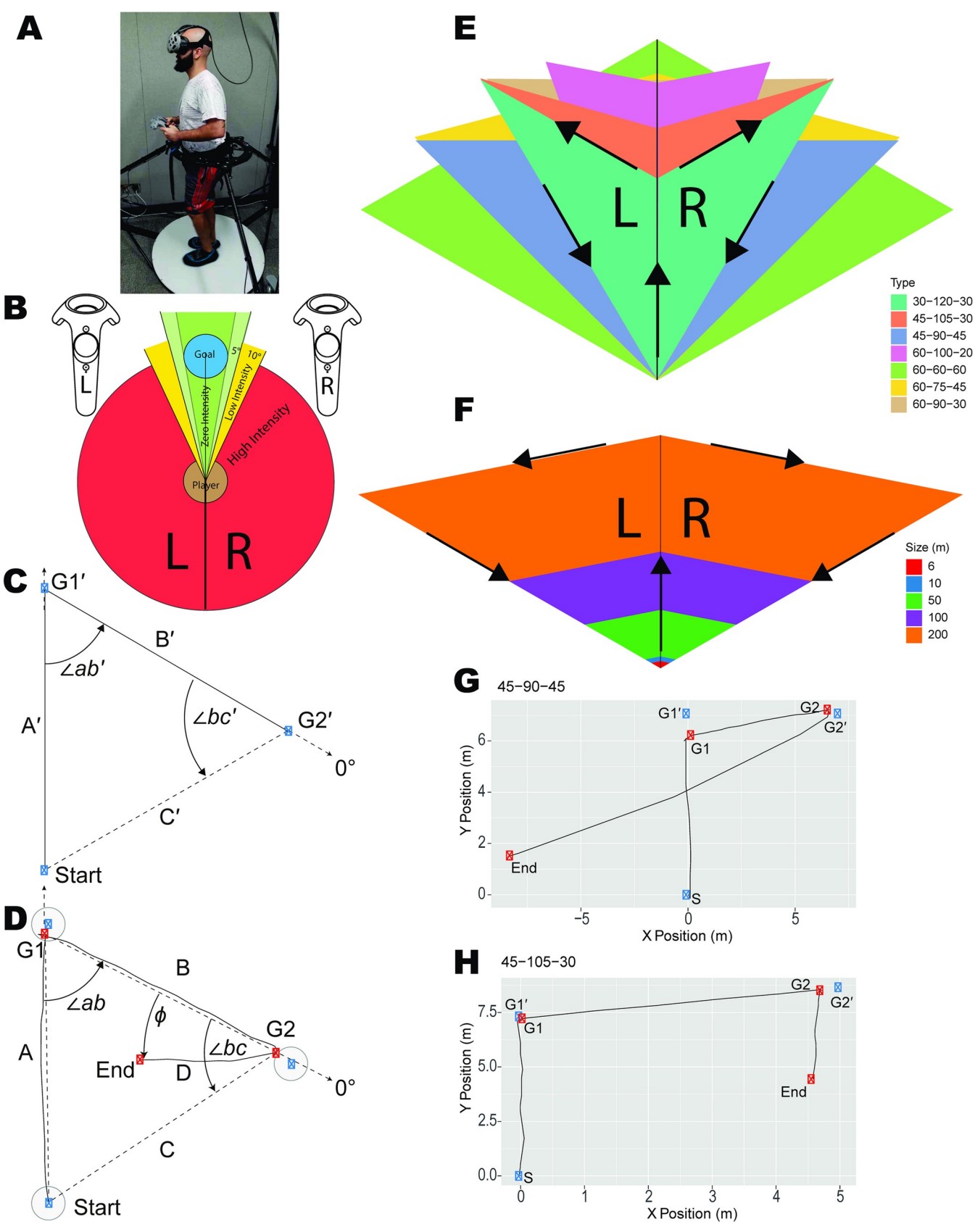

**Fig 1. Overview of experiments.** (A) HTC VIVE headset along with the handheld controllers used in the experiments. Participants walked on the Cyberith Virtualizer omnidirectional treadmill system allowing them to navigate a much larger space while in stationary ambulation. (B)Visualization of HTC VIVE handheld controllers' feedback intensity based on the deviation of the angle. (C) Depiction of an equilateral triangle used in experiment 1. (D) Raw trace of participant's path overlaid on the vector distances (dashed lines) between the points. The blue point denotes the G1' and G2' locations that the participant is guided to and the red points are the participant's unique G1 and G2 locations for that trial. "D" refers to the actual distance walked by the participant when attempting to return to the origin. (E) Triangle templates used in experiment 1 overlaid on top of each other showing the different triangle types and denoting the internal angles. (F) Triangle templates used in Experiment 2 overlaid on top of each other showing how the length of side C varied over trials. (G) Raw trial where the participant over estimated distance and the angle. (H) Raw trial where the participant underestimated the distance and the angle.

correct angle ($\angle bc$) and participant's heading angle for side D ($\phi$). This yields $\angle bc - \phi$ (Fig 1), in which a positive number denotes an overshoot and negative an undershoot. Distance error is the ratio of side D (the participants unguided walked distance) over the distance of C (configural homing vector from G2); a value greater than 1 is an overshoot and less than 1 an undershoot. As can be seen in the raw example shown (Fig 2A & 2B) and others (S1 Fig), although participants were often quite accurate at completing the triangle, they tended to overestimate the angle and underestimate the distance, regardless of triangle type. We will compare our findings of systematic errors with prior literature, specifically with Klatzky et al. 1999 [32] in the Discussion section.

**Participants overestimate angle and underestimate distance.** We next addressed the extent to which this overestimation of angle and underestimation of distance was true across the group of participants. As shown in Fig 2A, we found a tendency for participants to overestimate the angle they needed to turn to reach their start point (1-sample t-test against 0: t(21) = 3.7,p<0.001, Cohen's d = 0.79,$BF_{10}$>10), with participants, on average, tending to turn about 34.71˚±9.37˚ too far when estimating the angle they would need to turn to reach the origin. In contrast, we found that participants tended to underestimate the distance they needed to walk to get back to the start point, with participants' normalized walked distance significantly less than 1 (see Fig 2B, 1-sample t-test against 1: t(21) = 16,p<0.001, Cohen's d = 3.42, $BF_{10}$>10). The normalized walked distance was 0.87±0.05 (8.70m±0.50m). To determine the overall accuracy of the walked distance, we regressed the configural homing vector (side C) onto participants' unguided walked vector (side D). The beta values were positive and well above zero (1-sample t-test against 0: t(21) = 5.4,p < .001,Cohen's d = 1.151, $BF_{10}$>10), demonstrating that participants, despite underestimating distance, were well above chance in their estimates.

**Sensory modality of guidance information.** To ensure that our results were not due to difficulty with employing the handheld controllers to navigate the guided sides, we compared against a subset of trials in Experiment 1 in which the guided sides involved a visual beacon (note that participants otherwise navigated the unguided sides identically in somatosensory and vision conditions). During the *guided* section of the trials, there was no effect of vision (S2A Fig 2-sample t-test: t(21) = 1.09, p = 0.288, Cohen's d = 0.336 and $BF_{01}$>3), confirming that the handheld controller feedback system provides sufficient guidance. For angle error on the unguided side, as shown in S2B & S2D Fig, we found a slight but significant improvement in the vision-on (SD:43.10˚) compared to vision-off (SD:46.51˚) condition (2-way ANOVA with main effect of vision: F(1, 21) = 4.9, p<0.026, η2 = 0.016 $BF_{10}$ = 1.16). For distance error, as shown in S2C & S2E Fig, we also found a decrease in distance error during vision-on (SD:0.256) trials compared to vision-off (SD:0.271; 2-way ANOVA with main effect of vision: F(1, 21) = 8.2, p<0.004, η2 = 0.026 $BF_{10}$ = 4). We found no interaction effect between vision and triangle type. These findings suggest that providing vision on the guided sides did improve angle and distance estimates on the unguided side, but that participants still tended to overestimate angle and underestimate distance (see S2D & S2E Fig for additional information). Klatzky et al. 1999 [32] also found partial vision to improve accuracy, though, similarly, it seemed to have little effect on the direction of systematic errors. Thus, the overestimation of

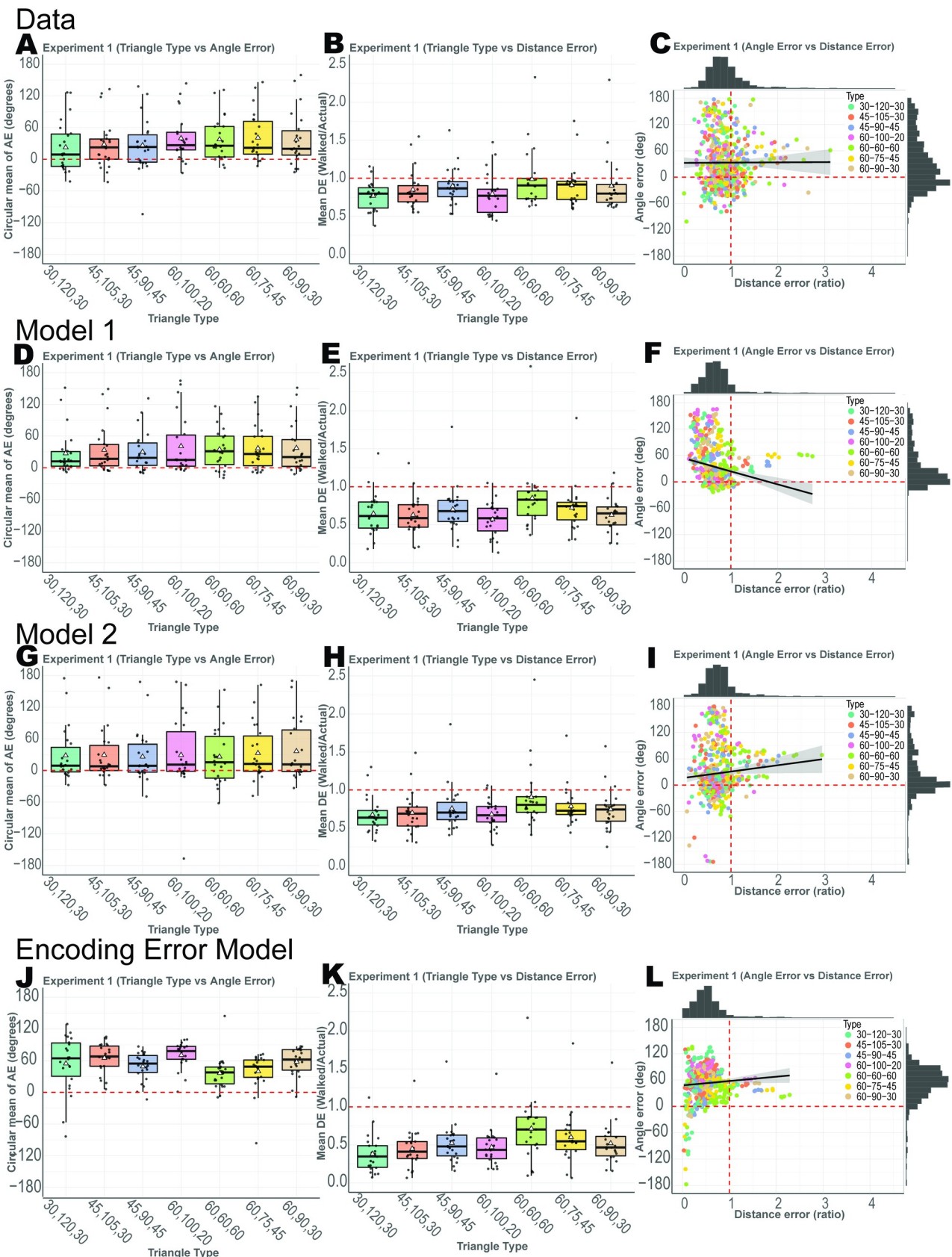

**Fig 2. Results from Experiment 1 along with model fits.** White triangles represent the mean, while the median is shown as a black bar. (A) Participants' circular mean of angle error for the 7 triangle types. (B) Participants' mean distance error for the unguided side. (C) Angle error and distance error for all trials. Model 1 simulated data from fitted values are shown as (D-F). Model 2 simulated data from fitted values are shown as (G-I). Encoding-Error model simulated data from fitted values are shown as (J-L).

angle and underestimation of distance that we observed cannot be accounted for by difficulty in completing the unguided sides using somatosensory input alone.

**Effect of triangle shape on patterns of errors in path integration.** Next, we wished to address the issue of triangle shape and whether this may have contributed in any way to the patterns of errors for the unguided side, as this might suggest participants used geometric features to anchor their path integration knowledge. For example, it could be that participants were most accurate for distance and angle on one triangle type (for example, right or equilateral triangles). To address this issue, we compared error on the unguided side with triangle type as an independent factor. Overall, we found only a modest effect of triangle type on angle error (2-way ANOVA with main effect of triangle type: $F_{(6,21)} = 2.9$, $p < 0.01$, $\eta^2 = 0.058$, $BF_{10} = 1.72$). Distance error, however, showed a fairly robust difference as a function of triangle type (2-way ANOVA with a main effect of triangle type: $F_{(6, 21)} = 5.7$, $p < 0.1.33e\text{-}5$, $\eta^2 = 0.109$ $BF_{10} > 10$); see Fig 2A and 2B. While we did not find a consistent effect of triangle type across angle and distance errors, it is clear the triangle type contributed significantly to the patterns of participant errors. For example, the isosceles triangle (30,120,30) showed the lowest mean angle error (10.96˚±9.11˚) yet the equilateral triangle demonstrated the lowest mean distance error (0.985±0.062). Whether participants were relying on triangle templates, abstract geometric cues, or instead repeating identical/similar past paths in working memory, is an issue we return to in the Discussion.

As an additional analysis to investigate the use of geometric features of triangles, if participants were using specific shapes over others to perform the task, we might expect that both angle and distance errors would be correlated. This would be consistent with using the shape, rather than individual features, to compute the unguided side. Comparing angle and distance error is also important in determining the extent to which these two estimates were stored in a common vs. independent manner. We found no correlation between angle and distance error across trials and participants (Fig 2C; $r(579) = 0.0035$, $p = 0.933$), suggesting that angle and distance errors were not related to each other. We also observed no clustering of angle and distance error by triangle type (Fig 2C). Finally, we looked at the left and right-handedness of the triangle and found no difference between them (S4A & S4B Fig; angle error 2-sample t-test: $t(21) = 0.7$, $p = 0.485$, Cohen's $d = 0.118$, $BF_{01} > 3$ and 2-sample t-test: distance error $t(21) = 1.136$, $p = 0.268$, Cohen's $d = 0.103$ and $BF_{01} = 2.53$). Together, these findings suggest that while geometric features (angle and distance) contribute to participant errors, it is not clear how these features are combined (if at all) to create and deploy triangle templates.

**Vector addition model.** To better understand the pattern of errors that participants made in Experiment 1, we built a configural computational model to predict the pattern of errors for the unguided sides. We combined angle and distance into a single vector value (see Methods) and employed the vectors for guided sides A and B as predictors for the unguided side. Based on previous findings [31], we would expect the guided sides to strongly predict performance on the unguided side. The modeling approach we employed also allowed us to compare the relative weighting of side A vs. side B and whether past trial history had any impact on unguided side performance.

We employed two versions of a simple vector addition model to fit our results. Model 1 assigns weights to sides A and B and predicts the distance and direction of side C (Eq 5). Model 2, in addition, includes a weighted influence of the participants' past trial history (Eq 6).

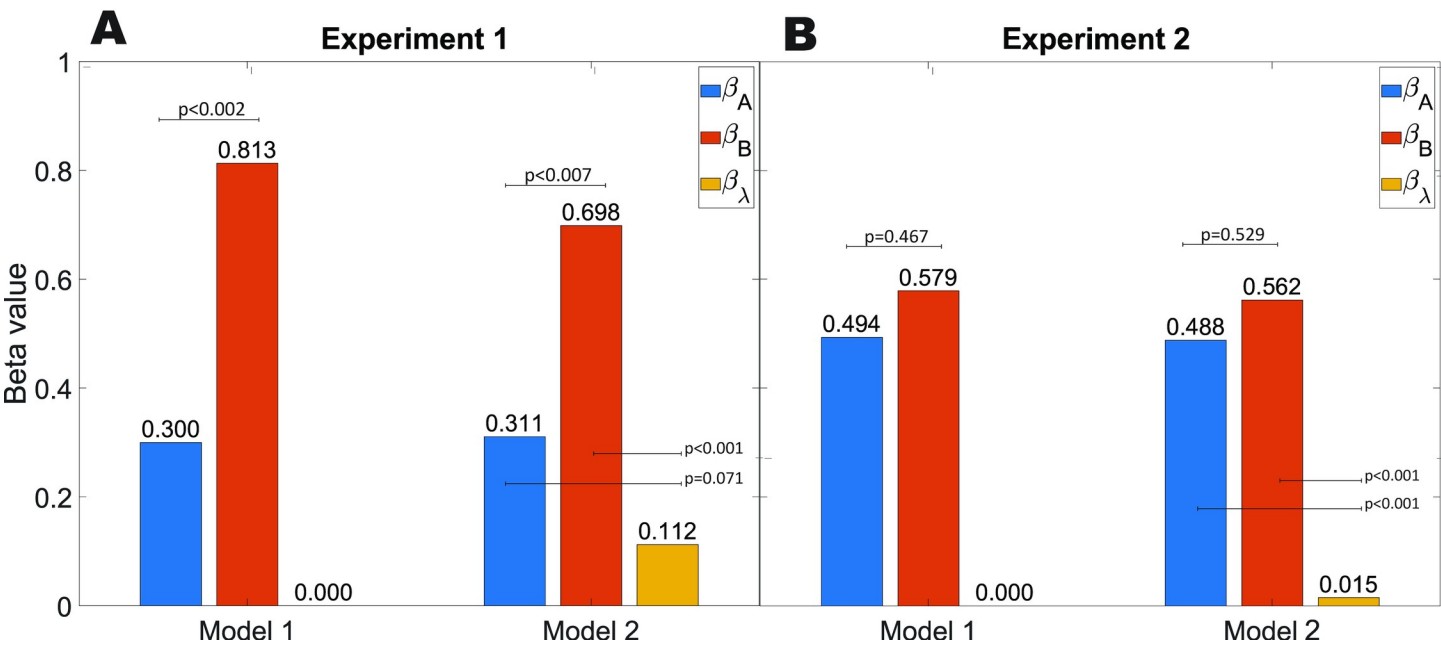

**Fig 3. Mean Beta values from the vector models.** (A): experiment 1 and (B): experiment 2.

We compared these two vector addition models to the Encoding-Error Model [31]. For Model 1, both guided sides A and B strongly predicted performance on the unguided side (mean $\beta_A$ = 0.3, 1-sample t-test against 0: t(21) = 2.86 p<0.0001 BF$_{10}$>3 and mean $\beta_B$ = 0.813, 1-sample t-test against 0: t(21) = 7.41 p<0.0001, BF$_{10}$>10; Fig 3A). Notably, only the beta values for side A were significantly less than 1 (1-sample t-test against 1: t(21) = 6.6, p<1.299e-6, BF$_{10}$>10), suggesting that participants underweighted side A when estimating the return vector. The underweighting of side A could, potentially, account for the angle overestimation. In addition, side B was weighted significantly higher than side A, (2-sample t-test: t(21) = 3.62, p<0.002, Cohen's d = 1.02, BF$_{10}$>10).

For model 2, which included participants' past trial history, we found mean $\beta_A$ = 0.311, 1-sample t-test against 0: t(21) = 3.05, p<0.006, BF$_{10}$ = 7.5 and mean $\beta_B$ = 0.698, 1-sample t-test against 0: t(21) = 7.775, p<0.0001, BF$_{10}$>10, suggesting similar results in terms of underweighting the guided sides as Model 1. We also found a significant effect of past trials (mean $\beta_\lambda$ = 0.112, 1-sample t-test against 0: t(21) = 3.415, p<0.003, BF$_{10}$>10), suggesting that sequential effects contributed significantly in Experiment 1 (Fig 3A). While the priors (past walked triangles) were similar in size in Experiment 1 (i.e., distance was not explicitly manipulated), the modeling results showed that past trials made a significant contribution to participant's estimation of the unguided side.

Taken together, these findings suggest that the findings from Experiment 1, which involved different triangle types, could be captured by our configural vector-based models, particularly Model 1. Participants underweighted both guided sides A and B, with a tendency to underweight side A to a greater extent, possibly accounting for the tendency of participants to overshoot the unguided angle. We also found evidence for a linear combination of past trials providing explanatory power for the unguided side.

**Model validation.** Next, we simulated Model 1 to determine whether it could account for the trends observed in the empirical data [47, 48]. We found that Model 1 captured both the angle overestimation (Fig 3A) and distance underestimation (Fig 3B) in Experiment 1. The

simulation results supported the idea that Model 1 provided a better account for the data than Model 2 (Fig 2D–2I) and captured the relevant empirical phenomenon reported here.

**Encoding-Error Model.** We fitted and simulated our data using the Encoding-Error Model, and, similar to Model 1 and Model 2, were able to capture the systematic errors in angle overestimation (Fig 2J) and distance underestimation (Fig 2K). The Encoding-Error Model, given the limited range of triangle distances in Experiment 1, did not show regression to the mean (effect of past trials). When we directly compared the models (S2J and S2K Fig), however, we found that Model 1 fit the data fairly decisively, at both subject and group level, compared to the other two models. While Model 1 did outperform the other two models in BIC and AIC, the confusion matrix in S7A–S7C Fig showed that simulated data from Encoding-Error model did not fit Encoding-Error model best compared to the two vector addition models. This method of model recovery suggests some limitations with our model comparison (i.e., how well our task can distinguish between models) and was likely due to small number of trials and the fact that the vector addition models involved far fewer free parameters than the Encoding-Error Model [48]. We return to a more detailed comparison between vector addition and Encoding-Error Models in the Discussion.

## Experiment 2

**Basic behavior.** In Experiment 2, we manipulated the distance of the triangles (perimeters = 15.19, 25.32, 126.60, 253.20, and 506.42 meters) while keeping triangle geometry constant. This involved manipulating the distance of the guided sides while maintaining a scalene triangle shape, thus leaving the angles constant. We implemented the same task structure as Experiment 1 but here we kept the shape of the guided path the same and varied the scale across trials.

**Participants systematically underestimated distance but accurately estimated angle.** For angle error, somewhat in contrast to Experiment 1, we found no significant overestimation or underestimation of angle, with participants showing a mean error of $0.8°\pm7.44°$ (1-sample t-test against 0: $t(16) = 0.107$, $p = 0.916$, Cohen's $d = 0.026$, $BF_{01} > 3$). We also found no effect of triangle size on angle error (Fig 4A 1-way ANOVA with no main effect of triangle size: $F(4,16) = 0.609$, $p = 0.658$, $\eta^2 = 0.036$, $BF_{01} > 10$). We attribute this to the fact that triangle configuration was consistent across Experiment 2 because we primarily manipulated distance.

We found evidence of fairly accurate estimation of distance for smaller triangle perimeters (15-25m perimeter) and considerable underestimation for larger triangle perimeters (126m – 500m perimeter). In fact, we found a trend whereby distance underestimation increased as a function of the unguided distance (Fig 4B, 1-way ANOVA with main effect of triangle size: $F(4,16) = 21.107$, $p < 3.913e-11$, $\eta^2 = 0.553$ and $BF_{10} > 10$). This is shown in Fig 5A, where the dotted line indicates a slope of 1, with the actual slope well below this value. In other words, the further that participants walked, the more they tended to underestimate the unguided side.

To better understand this phenomenon, we analyzed the spread of the errors as participants walked the unguided side. We found that the distribution of distance error scaled linearly as a function of the walked distance. As shown in Fig 5B, the standard deviation of the walked unguided distances increased linearly, as shown by a regression fit (linear model: $t(4) = 23.6$, $p < 0.0001$), suggesting that the greater the walked distance, the proportionately greater the error in distance with variance increasing exponentially. Distance underestimation $(1 - \frac{side\ D}{side\ C})$ increased as a function of distance as well, however, this increase was best fit by a logarithmic function (Fig 5C, linear model: $t(4) = 13.39$, $p < 0.001$) rather than linearly, similar to Weber–Fechner and Stevens' power law [49]. Together, these findings suggest that as participants

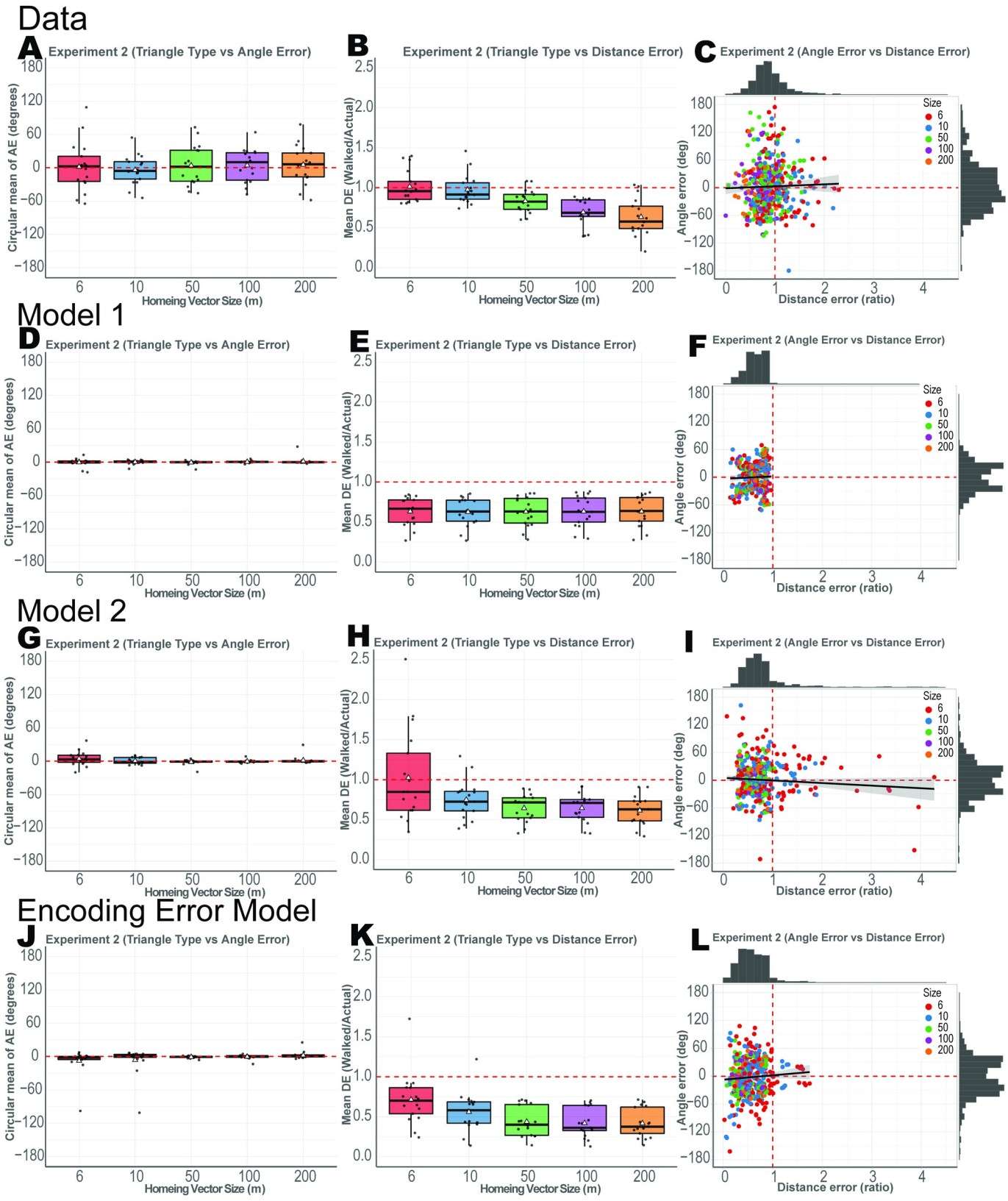

**Fig 4. Results from Experiment 2 along with model fits.** White triangles represent the mean while the median is shown as a black bar. (A) Participants' circular mean of angle error for the 7 triangle types. (B)Participants' mean distance error for the unguided size. (C)Angle error and Distance error of all trials. Model 1 simulated data from fitted values are shown as (D–F). Model 2 simulated data from fitted values are shown as (G–I). Encoding–Error model simulated data from fitted values are shown as (J–L).

walked longer distances, they tended to increase their underestimation of the distance they would need to walk and scale their errors logarithmically as a function of distance.

Similar to Experiment 1, we also found no correlation between angle and distance error (Fig 4C, 2-sample t-test: t(487) = 0.623,p = 0.533, $BF_{01}$>7.8). We also found no effect of right vs. left turns on guided sides (angle error: 2-sample t-test: t(16) = 1.51, p = 0.151, Cohen's d = 0.245, $BF_{01}$ = 1.55 and distance error: (2-sample t-test: t(16) = 0.724, p = 0.4797, Cohen's d = 0.176 and $BF_{01}$ = 3.188), see S4C & S4D Fig.

**Vector addition model.** To better understand the effects of the guided sides on the unguided side estimates in Experiment 2, we employed the computational models used in

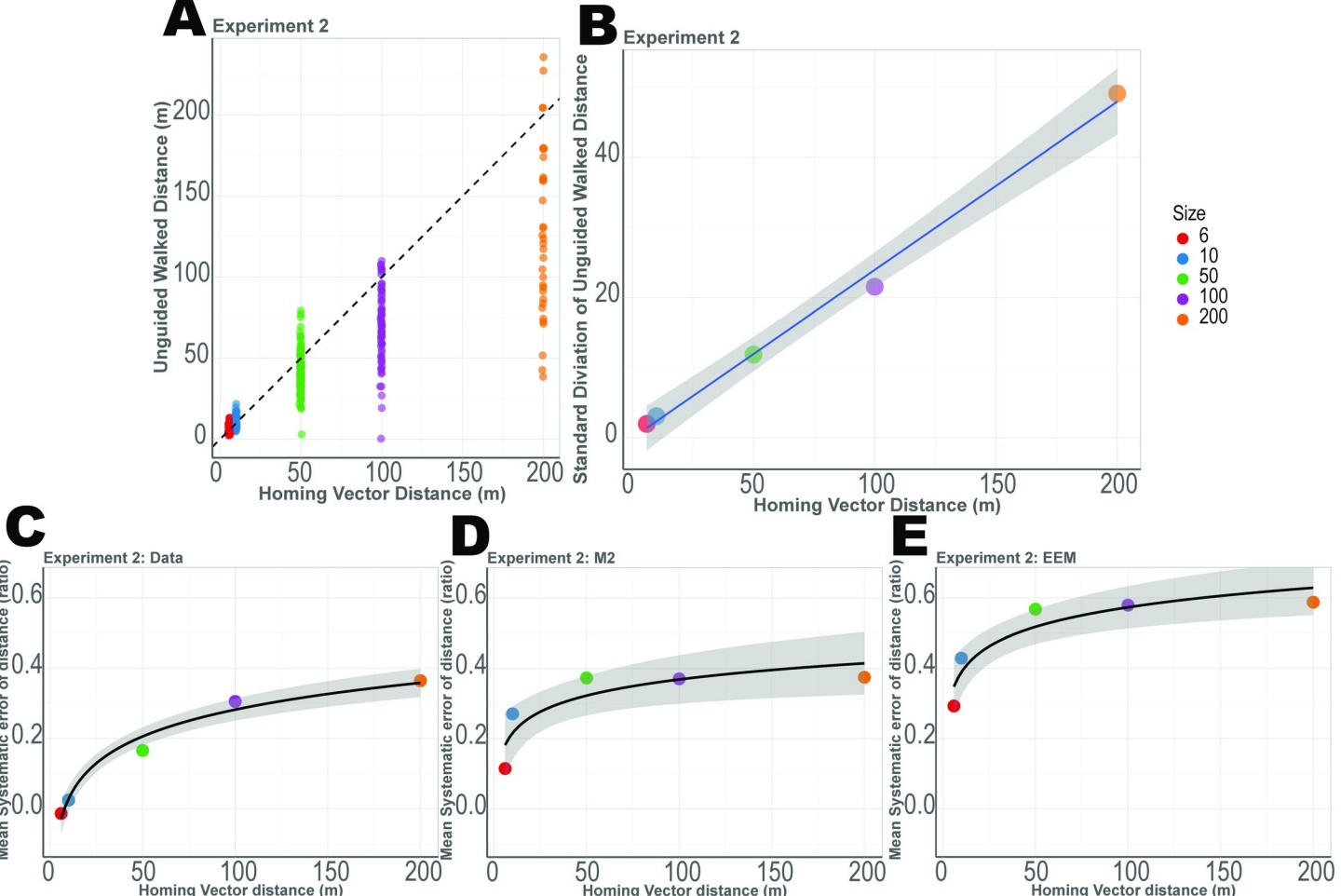

**Fig 5. Systematic underestimation of distance from Experiment 2.** (A) The distribution of distances for the unguided side (D) for each triangle size with y = x plotted at the dotted line. (B) Standard deviation of distances for the unguided side (D), which show a linear increase. (C) Mean systematic errors of distance (1- distance error), which increase logarithmically. (D) Mean systematic errors of distance of the simulated data from Model 2, which increase logarithmically. (E) Mean systematic errors of distance of the simulated data from Encoding-Error Model, which increase logarithmically.

Experiment 1 to predict the pattern of errors for the unguided sides. The modeling analysis from Model 1 again revealed that both guided sides A and B strongly predicted performance on the unguided side (mean $\beta_A$ = 0. 494, 1-sample t-test against 0: t(16) = 5.09, p<0.001, $BF_{10}$>10 and mean $\beta_B$ = 0. 579, 1-sample t-test against 0: t(16) = 9.33, p<0.001, $BF_{10}$>10; Fig 3B). Notably, both beta values were less than 1 (1-sample t-test against 1: $\beta_A$ t(16) = 5.22 p<0.001, $BF_{10}$>10 and 1-sample t-test against 1: $\beta_B$ t(16) = 6.80 p<0.001, $BF_{10}$>10), suggesting that participants underweighted *both* sides when estimating the return vector. In addition, unlike Experiment 1, both sides were weighted evenly (1-sample t-test: t(16) = 0.547,p = 0.591, Cohen's d = 0.218, $BF_{01}$>3). These findings are perhaps unsurprising because angle was neither under nor overestimated.

Comparing Model 1 (modeling the distance of the guided sides to predict the unguided sides) and 2 (using Model 1 with an additional term for past trial distances), we found significant fits for all three beta terms. In other words, guided sides A & B, as well as past trial history (mean $\beta_A$ = 0. 488, 1-sample t-test against 0: t(16) = 5.06, p<0.001, $BF_{10}$>10, mean $\beta_B$ = 0. 562, 1-sample t-test against 0: t(16) = 8.96, p<0.001, $BF_{10}$>10 and mean $\beta_\lambda$ = 0.015, 1-sample t-test against 0: t(16) = 3.26, p<0.005, $BF_{10}$ = 9.83), all predicted errors in walking the unguided side in Experiment 2. Thus, similar to Experiment 1, trial history provided a significant explanation of error in Experiment 2. Note, $\beta_\lambda$ values are smaller in Experiment 2 due to the large values of $\lambda$ (linear combination of past paths), which in Experiment 2 can go up to 500m.

**Model validation.** Next, we simulated our data in a manner similar to Experiment 1. Simulated data from Model 1 showed that we were able to capture participant patterns in angle error (Fig 4D). While Model 1 captured the distance underestimation (Fig 4E), it did not capture the trend of increase in underestimation as a function of distance. We hypothesized that this effect could be an influence of past trials, in other words, a form of regression to the mean [35, 36]. Fig 4G shows the simulated angle error from Model 2. We were again able to capture the accurate angle predictions. Importantly, however, simulated distance error for Model 2, as shown in Fig 4H, better captured the pattern of distance underestimation. Model 2, in particular, captured the tendency of participant underestimation of distance to increase as a function of distance while Model 1 (which did not include trial history) was not able to capture this effect. These findings suggest that the increasing underestimation of distance was influenced, in part, by past trials. This effect was likely stronger in Experiment 2 than 1 due to the longer range of distances employed.

**Encoding-Error Model.** The Encoding-Error Model also captured some of the same patterns in the data as Model 1 and 2. The simulated data from the Encoding-Error Model showed accurate angle error and underestimation of distance errors as a function of distance (Fig 4J & 4K). We also considered how well the Encoding-Error Model compared with Model 2 in terms of capturing the mean systematic error in distance, which was 1- mean distance error (Fig 5C–5E). While the Encoding-Error Model fit the logarithmic function of systematic errors, the values were less accurate than Model 2. Similar to Experiment 1, Model 2 best fit the data, but the BIC and AIC favored Model 1 (S6D–S6F Fig). Notably, though, our analyses (see Fig 2D–2F) suggested that Model 1 did not capture the pattern of systematic errors (distance undershoot) and thus we removed it from the model comparison with the Encoding-Error Model. As shown in S8A–S8D Fig, we can see Model 2 fits 11 subject's data better while the Encoding-Error Model fit the other 5 subjects' data better. Similar to Experiment 1, the confusion matrix (S3A–S3C Fig) showed that Encoding-Error model did not fit its own simulated data well. This was likely due to small number of trials and the fact that the vector addition models involved fewer free parameters than the Encoding-Error Model (S1 Text). We return to a more detailed comparison of the models in the Discussion.

## Discussion

In two different experiments, participants were guided on two sides of a triangle and then attempted to return to the origin without any input using a novel interface involving an omni-directional treadmill. In Experiment 1, we manipulated triangle type (equilateral vs. isosceles vs. right vs. scalene) while holding distance on the unguided side constant to minimize prior effects. Consistent with previous work using the triangle completion task in small-scale room-sized environments [21, 31, 32, 50, 51], we found that participants underestimated distance and overestimated angle. These systematic errors, however, did not show a regression to the mean effect. In Experiment 1, our computational modeling results suggested that this pattern could be explained by a model in which participants underweighted side A compared to side B with an effect of past trial history. In Experiment 2, we found systematic errors in distance as participants accurately estimated the angle they needed to turn while increasingly underestimating the unguided side as a function of distance, consistent with logarithmic scaling described in the Weber-Fechner law. Modeling results for Experiment 2 further suggested equal weighting of both encoded sides. We also found no correlation between angle and distance errors in both experiments, consistent with reports that, at least in part, we derive angular motion from the semicircular canals and linear motion through the otoliths [52]. Our findings thus suggest that participants use independent estimates of direction and distance to estimate a configural homing vector, with the current trial guided sides influencing estimates of the configural homing vector.

In Experiment 1, we found that the triangle type tested had an influence on participants' distance and angle errors on the unguided side. We found that the equilateral triangles showed significantly lower distance error and similarly, we found a weak tendency for the isosceles triangle angles (30,120,30) to show lower angle overestimation. All triangle types observed in Experiment 1, however, showed an angle overestimation and distance underestimation. The question of whether the differences in distance and angle errors observed are due to participants using triangle templates is one we sought to answer in this study.

We believe that If participants were using a triangle template, then we would be able to observe accurate distance *and* angle estimations for one triangle type. For example, it might be possible to predict that equilateral triangles or right triangles would be overall more accurate than scalene triangles. This is because these geometries are far more regular and potentially easier to encode holistically, particularly given their influence on visually-guided navigation [53]. While we did not find one triangle type to have lower distance *and* angle errors at the group level, there may be several explanations for how triangle type had an influence on participants' performance.

First, we found two participants who showed a high degree of accuracy for both distance and angle on equilateral and right isosceles triangles (S3C Fig). This suggests that some participants may have used a strategy that involved deploying templates. Unlike Wiener et al. 2011 [34], we did not explicitly ask participants to focus on the outbound path or the origin, which in turn could possibly have affected some participants' likelihood of spontaneously using a template. Our results, in this way, build on other studies showing individual differences in human navigation studies [54–56].

A second explanation for how triangle type may have influenced participants' performance focuses on the possibility of using abstract geometric features. Studies in young children suggest that rather than using room geometry to navigate, children use certain geometric features (i.e., the distance between walls) to reorient themselves [57, 58]. In our task, participants were not directly exposed to the geometric regularities of the path but rather they had to "trace" it by walking it. This process may conserve some geometric features and not others, which may

in turn influence the pattern of errors. In particular, because participants could not directly perceive the entire shape, they may have attempted to guess it, and if their memory for a side was inaccurate, this would affect their memory for the entire shape. Thus, while we could not determine any correlation between patterns of errors and any obvious geometric features, there may be other features that were not detectable for our experimental task. These could have been influenced by geometry although in a way not detectable in our design due to tracing the path over several minutes rather than directly perceiving it in one glance.

Third, it may be that participants simply do not rely on any geometric features. Instead, it is possible that the small distance errors for the equilateral triangle type can be attributed to a working memory effect based on the equivalence of all three side distances. The overall systematic trends of overestimating angle and underestimating distance may have little to do with encoding geometric properties and instead be a result of under-encoding the guided legs and execution errors [33, 59]. Overall, the lack of any consistent effects in angle and distance for specific triangle types in terms of accuracy and the lack of a correlation between angle and distance suggest, on a group level, that participants may not have been using triangle templates. The fact that triangle type does have separate yet significant influences on individual distance and angle errors leaves room for further investigation.

In Experiment 2, we tested path integration over distances much longer than those typically employed in past human studies. Almost all of our current knowledge base about path integration mechanisms in humans derive from testing in room-sized environments, and therefore, in contrast to what is known about other species, the extent to which path integration mechanisms operate accurately over distances greater than 10 meters remains unclear. We found that participants were fairly accurate in their ability to complete the third side of a triangle, even for triangle perimeters as long as 500 meters. Although we found a systematic increase in error and underestimation as a function of longer distances, these biases increased logarithmically, suggesting that the basic mechanisms underlying path integration were not substantially different at 500 meters compared to 25 meters. In contrast to Experiment 1, we found that both sides A and B contributed equally to errors in unguided side C, although we attribute this effect to the fact that we did not manipulate angle in Experiment 2. We found that past trial history contributed significantly to the pattern of errors at longer distances. These findings suggest that in fact some of the properties of path integration do change somewhat over longer distances, particularly the tendency to erroneously weigh past trials to estimate the current ones. Given that our two models, however, involved the same basic conceptual setup (side A +B = C), these results suggest that the basic mechanism of adding vector values for the guided sides to compute a configural homing vector held constant across experiments.

Our computational modeling results indicated an effect of past trials on participant error patterns in both experiments. In other words, for the longer distance triangles, we found a weak, but significant bias for past trials to influence the extent to which participants underestimated the amount they needed to walk on the current triangle. For large triangles, therefore, shorter past trials would result in a greater tendency to underestimate distance. Notably, including the history term in our model significantly increased our ability to account for the increasing tendency of participants to undershoot the distance they needed to walk on the unguided side. These findings support the idea that for particularly long distances, path integration is also influenced by a form of regression to the mean from past trials, thus explaining why undershoot increased with longer distances. These findings, which, to the best of our knowledge, have not been demonstrated previously at such long distances in humans, suggest that path integration is not merely a function of the current walked triangle, but is also influenced by the memories and experiences of past trajectories.

Because of our strong reliance on visual input, testing humans in the absence of vision is challenging, particularly due to the possibility of trip hazards and collisions. Thus, many researchers have chosen to investigate path integration using desktop VR, which also allows simultaneous brain imaging, for example, using fMRI [54, 60]. One limitation with desktop VR is that although it preserves optic flow, it lacks the rich cues that one obtains from freely moving the body in space [19]. These include vestibular information from head turns, proprioceptive information about body position, efferent copy from motor movements, and somatosensory input from the feet as they move over the surface [16] [61–66]. Our novel interface was able to reproduce many of these cues, particularly those that would be expected from angular vestibular cues, turning, and shuffling the sides of the feet. The treadmill, however, has diminished translational vestibular cues although it is notable that the distance underestimation results we obtained were largely comparable to past studies using the real-world triangle completion task [32]. As such, we were able to capture novel aspects of non-visual navigation otherwise difficult to observe in desktop-VR.

Additionally, participants in our study generated their linear and angular motion, while non-VR versions of the triangle completion task used in the past relied on the experimenter physically guiding the participants' movements. Previous versions of path completion tasks have used an object (rod or rope) in which the experimenter guides the participants by pulling or lowering for turning [21, 32, 35]. In contrast, in our design, participants received feedback from handheld controllers indicating which way to go. We believe that the use of feedback via handheld controllers, rather than external forces to guide participants, better approximates active walking. Specifically, active walking requires one to initiate the movement while outside forces that initiate or guide the movement would typically be referred to as passive. We believe by controlling for active walking during the guided portion, we have better controlled for differences between guided and unguided conditions. While the distinction between active and passive movement is a subtle one, recent work suggests important differences between these two forms of walking in terms of their neural bases [52].

## Model comparisons

Vector addition has long been assumed to be the functioning principle for path integration [23, 24, 26]. The vector addition models proposed in this paper (Models 1&2) assume that the configural homing vector is computed by summing vector representations of sides A and B. In contrast, the Encoding-Error Model assumes that the configural homing vector is computed using the distance and angle values experienced during the entire guided portion. While both models are similar in aim, we believe the computational principles for the vector model may be more plausible. To employ the Encoding-Error Model, participants must form a representation of the linear relationship between distance guided and distance walked (distance representation) as well as for turns, for each path configuration. In contrast, vector addition models assume a linear relationship between the guided sides and the configural homing vector, with the possibility of prior trials influencing the current trajectory.

As mentioned in the Introduction, there are other reasons to think that vector addition models confer advantages, particularly in accounting for human path integration findings from the triangle completion task. The Encoding-Error Model has four requirements, with one important assumption being that the internal representations must obey Euclidean axioms. Recent papers, however, suggest that human spatial navigation, in some instances, may be better characterized by representations based on non-Euclidean labeled graphs [67]. Specifically, Warren et al. 2019 [67] described path integration using simple vector manipulations with such manipulations preserved in non-Euclidean spaces. Our model, which can be readily

adapted to non-Euclidean geometries, would therefore also provide greater flexibility than the Encoding-Error Model in terms of fitting violations of Euclidean axioms.

Another requirement of the Encoding-Error Model is the assumption that all systematic errors occur during encoding rather than during spatial reasoning or execution. Vector addition models are more flexible, assuming systematic errors can aggregate at different stages, whether it is during encoding, retrieval or computation of the configural homing vector. The Encoding-Error Model, however, is limited in that side A and B derive from the same linear function, such that side A cannot be underestimated more than side B. There may be instances, however, in which one side is weighted differently in a path with 2 segments compared to 5 segments [59]. This was also true in Experiment 1, where we found that side A was under-weighted compared to side B.

In addition, the Encoding-Error Model is limited to 2 segmented triangular paths, based on the law of cosines (S1 Text) and does not perform well with 3 segmented paths [31]. In contrast, vector addition models can readily be extended to n paths with the caveat of adding a free parameter with each segment. Notably, the vector addition models we employed here provided an overall better fit of the actual data (S6A & S6D Fig). We note, though, that the Encoding-Error model cannot be fully differentiated during model recovery (S7 Fig). The likely reason for this is the small number of trials in our task. While both Model 2 and Encoding-Error model can account for some patterns in the data, including systematic errors, importantly, Model 2 has the best log-likelihood fit (S6A & S6D Fig), despite the Encoding-Error model having more free-parameters. Overall, therefore, we think the vector addition models provide a better fit of our data and are more parsimonious although more work is needed to allow a detailed and formal model comparison.

## Limitations of the Vector Models

While the vector addition models employed here do a fairly good job of capturing the patterns of our findings in the two experiments in this study, they are not without certain limitations. One issue is that the model in its current form assumes that side A and side B are encoded with similar directions (i.e. $\beta_A x_A^t$ has the same direction as $x_A^t$) or opposite directions (S10 Fig). Our current versions also assume that only the vector magnitudes affect systematic errors (see S10 Fig). We hope to address the issue of vector directions in more detail in future models.

## Methods

### Ethics statement

All participants gave written informed consent to participate in the study, which was approved by the Institutional Review Board at the University of California, Davis (Protocol: 816949).

### Training and the triangle completion task

We employed a task used previously to investigate human path integration termed the triangle completion task [21]. Briefly, the task involves guiding participants on two sides of a triangle and then completing the third side without guidance or feedback. Based on our goal of studying a variety of different triangle types and sizes, we adapted the task to an omnidirectional treadmill, the Cyberith Virtualizer treadmill. The task involved participants walking on the treadmill, with guidance on two of the sides provided by somatosensory feedback from HTC VIVE handheld controllers. Participants wore the HTC VIVE headmounted headset to allow us to track head and body position, as well as to limit visual input.

To first ensure that participants could walk comfortably in the treadmill, we employed a pre-experimental training session. We employed an HTC VIVE head-mounted display to give visual feedback to ensure balance and comfort on the treadmill. In the first part of the training, we included a 3-stage puzzle game created in Unity 2017.1.1f1 in which participants had to explore an environment to find an object. Once participants completed the 3-stage puzzle game, reported no cybersickness, and the experimenter determined that their walking technique was adequate, they advanced to the next level. At this point, we introduced the HTC VIVE handheld controllers feedback system (Fig 1B) and had subjects walk straight lines with no visual information while receiving feedback from the handheld controllers. This insured that they could accurately perform the guided sides. Following this, they performed a small number of practice triangles. After practicing the triangle completion task on 6 unique triangles, which were not included in the experiment, the experiment started. The training period ranged from 30–60 min. To ensure participant safety, we occasionally questioned them about how they were feeling to guard against issues with cybersickness.

Participants then proceeded to the main experiment. The first experiment involved manipulating triangle geometry (i.e., primarily the angles they turned) and Experiment 2 involved manipulating triangle size (i.e., we manipulated the distance they walked on the third / unguided side). Trial sequences were randomly chosen from 5 pseudorandomized configurations. In both experiments, we guided participants along the first two sides of the triangle using the handheld controllers' feedback system (Fig 1B). The feedback system was designed such that if the participants strayed from their path, the controller vibrated accordingly (for example if the right controller vibrated, they would need to turn left) to help guide them in walking in a straight line. When participants walked in the correct direction, the controller did not send feedback, allowing for active walking (passive guidance). Participants were guided alongside A' and then alongside B' by controller feedback (Fig 1C). At G2', the handheld controller feedback system turned off and participants were instructed to find their way to the start point. Participants pressed the trigger on the handheld controllers once they believed that they reached the start point. We constructed trial specific vectors to capture the performance variability during guided sides (see Fig 1E). We manually inspected these trials, and those which showed a clear deviation from linearity were excluded, which resulted in approximately 16.5% ($\frac{117}{696}$) of removal of trials from Experiment 1 and 6.88% ($\frac{36}{523}$) from Experiment 2 across participants. Participant data that exceeded 25% removed trials were excluded from the analysis. We redid the analysis by including all trials and participants and obtained similar results to what are reported here.

## Modeling

**Description of models.**   To further understand how the guided sides contributed to the angle and distance errors of the unguided side, we created a vector model of path integration. In this model, we assume that participants estimate a "configural homing vector", $x_C^t$, by combining the vectors corresponding to each of the guided sides for that trial (denoted by superscript t), $x_A^t$ and $x_B^t$. If path integration were optimal, people would combine these vectors in the following way

$$x_C^t = -(x_A^t + x_B^t) \qquad\qquad 1$$

and would return perfectly to the point of origin by walking along the vector $x_C^t$.

We assumed that people could over, or underweight, a given side when computing the sum–perhaps because they integrate evidence unevenly over time [68]. To model participants'

biased responses, we allowed $\tilde{x}_D^t$ to be a *weighted* sum of the vectors from the first two sides:

$$\tilde{x}_D^t = -(\beta_A x_A^t + \beta_B x_B^t) \tag{2}$$

Where $\beta_A$ and $\beta_B$ denote the weights given to side A and side B respectively (Fig 1D).

Of course, real participants are suboptimal and we modeled these suboptimalities in a number of different ways. First, people may not perfectly encode the vectors from the guided sides and/or may not perfectly implement the desired action, adding noise to the sum in Eq 2. Thus, we assumed that the vector they actually walked $x_D^t$ was sampled from a Gaussian distribution centered on $\tilde{x}_D^t$, i.e.

$$P(x_D^t|\tilde{\sigma}, \beta_A, \beta_B) = \frac{1}{\sqrt{2\pi\sigma^2}} exp(-\frac{(x_D^t - \tilde{x}_D^t)^2}{2\sigma^2}) \tag{3}$$

Where $\sigma^2$ is the variance of the noise. Consistent with Weber's law, we assumed this variance increased with the distance walked to match our finding of increased variance as a factor of distance walked in Experiment 2 (see Results, Fig 5C).

$$\sigma = \tilde{\sigma} * \sqrt{x_A^{t\,2} + x_B^{t\,2}} \tag{4}$$

Combining the first two sides gives us Model 1 (Eq 5), which includes noise and the possibility of over and underweighting the sides.

$$x_D^t = -(\beta_A x_A^t + \beta_B x_B^t) + \varepsilon \tag{5}$$

Where $\varepsilon$ is the noise term sampled from a normal destitution with mean 0 and standard deviation $\sigma$.

Finally, we allowed for the possibility that there may be sequential effects in our paradigm, i.e. there was an influence of previous trials on the current response. We modeled these sequential effects by including the vectors walked ($x_A^{t-n}$, $x_B^{t-n}$ and $x_D^{t-n}$) from past trials. For simplicity, we assumed that the effect of past trials decayed exponentially into the past [69], thus writing $x_D^t$ as

$$x_D^t = -(\beta_A x_A^t + \beta_B x_B^t + \beta_\lambda[\lambda_A^{t-1} + \lambda_B^{t-1} - \lambda_D^{t-1}])) + \varepsilon \tag{6}$$

$$\lambda_n^t = x_n^t + \alpha\lambda_n^{t-1} \tag{7}$$

Where $\lambda_n$ is a linear combination of the previous vectors, fitted with $\alpha$, which ranges between 0 to 1, to capture the impact of prior trials. Thus, including the possible effect of past trials gave us Model 2.

**Fitting the model.** We fit the model using a maximum likelihood approach. In particular, we computed the log likelihood of the responses for each subject, as a function of model parameters:

$$LL(\tilde{\sigma}, \beta_A, \beta_B, \beta_\lambda, \alpha) = \sum_{t=1}^{T} \frac{log(2\pi\sigma^2)}{2} \frac{(\tilde{x}_D^t - x_D^t)^2}{4\sigma^2} \tag{8}$$

We then found the parameters that maximized the likelihood using Matlab's fmincon function.

**Simulating the models.** To simulate the model, we used the parameter values fit for each subject to compute the mean $x_D^t$ for each trial. To model the noise in each person's choice, we perturbed the estimate of $\tilde{x}_D^t$ by isotropic Gaussian noise of mean 0 and variance $\sigma^2$.

**Encoding-Error Model.** We recreated the Encoding-Error Model from Fujita et al. 1993. See S1 Text for more details. We used the same fitting and simulation method used for Model 1 and Model 2 with the exception of dividing the data for left and right-handed triangle to better accommodate the parameters of the Encoding-Error Model [32].

**Model Comparison Methods.** We used two methods of model comparisons: 1) Penalized-Log-likelihood criteria's Bayes Information Criterion (BIC) [70] and Akaike information criterion (AIC) [71]. Both express similar information about the generalizability of the model by penalizing for the number of free parameters. To test how meaningful our model comparisons results are in our task we also tested for model recovery. We did this by simulating each model with randomized parameter values and then fitting the models to the simulated data, allowing comparison of the AIC and BIC (see section 6 and Appendix B in [48]). We performed each simulation at the participant level and then subsequently compared BIC values by calculating exceedance probabilities, which measured how likely it is that the given model fits all of the data [72]. This group level statistic is similar to AIC and BIC. Computed exceedance probabilities on our data as well as each model by simulating 100 times and comparing with the methods mentioned above. These methods are illustrated in S6 Fig where the probability of the model fit for the simulated data ranges from 0 to1. The Exceedance Probability is calculated using SPM 12 spm_BMS function.

**Bayes Factor Analyses.** We included a Bayes Factor analysis for all statistical analyses [73]. For results below our significance threshold ($p < 0.05$), we used a Bayes Factor $BF_{10}$ to indicate the degree of favorability toward the alternative hypothesis. For results that were not below our significance threshold, we employed the Bayes Null factor, $BF_{01}$. Note that the larger the Bayes Factor, regardless of whether in favor of the alternative or null, the greater the evidence.

## Experiment 1

**Participants.** We tested a total of 26 participants (12m,14f), 4 (1m, 3f) of which were removed due to exceeding 25% of trials removed (see methods), Participants were tested on 7 different triangles described in detail in the methods (i.e., scalene, isosceles, right, equilateral, and isosceles-right). Estimates of sample size were based on the 12 participants used in Loomis et al. 1993 and in subsequent studies by [74]. that employed a similar experimental design: as we were additionally testing a larger range of triangles, we thus approximately doubled the sample size.

**Procedure.** We outline the basic set up for triangle geometry in Fig 1E, which shows the stacked triangle templates, with a constant 10m side C' (unguided side), while manipulating the angle. The 7 triangle configurations are shown in S1 Table, with 3 scalene, 1 isosceles, 1 right, 1 equilateral, and 1 isosceles-right. To keep side C' at a constant 10m across all 7 triangles, we employed different side A' and side B' (guided portion) sizes to accommodate the different angles. Note side D is the participant's response, and distance errors are calculated as the ratio (side D) / (side C). There were 28 trials, in which 14 of them were left-handed (subjects only made left turn) and 14 right handed (subject only made right turns). We did this to avoid any advantages for right vs. left turns during the task.

In Experiment 1, as part of ensuring the compliance and efficacy of the handheld controllers in following the guided sides, we compared with a condition in which participants walked the guided sides on half the trials using a visual beacon. In this situation, participants saw a large red monolith that they walked to while receiving feedback from the handheld controllers. It is important to emphasize that the vision-guided trials were only present for the *guided* sides and were simply to ensure that participants accurately encoded the guided sides before performing the unguided sides.

### Experiment 2

**Participants.**   We tested a total of 21 participants (9m,11f), 3 (1m 2f) of which did not complete the experiment, with additional 1 female participant removed from the analysis for exceeding 25% trials below criterial performance. Given the longer distances in Experiment 2, participants were allowed to take a break, but only at the end of a trial. About 50% of participants took a break at some point during the experiment.

**Procedure.**   Here, we employed scalene triangles with different length perimeters to allow us to manipulate distance while keeping angle constant, testing 5 different triangle sizes. Fig 1F shows the stacked triangle templates we employed with constant internal angles but varying in size. The triangle configurations are shown in S2 Table, with 15m, 25m, 127m, 253m, and 506m perimeters. There were 30 trials, with 15 of them left-handed (participants only made left turn) and 15 right-handed (participants only made right turns). Unlike Experiment 1, there were no vision trials. Due to testing longer distances and wanting to avoid fatigue, we limited the number of trials for the longest distance triangles. The distributions of trials were 10 for the 15m triangle, 10 for the 25m triangle, 8 for the 127m triangle, 4 for the 253m triangle, and 2 for the 506m triangle.

## Supporting information

**S1 Table. The configuration of each triangle used in experiment 1.**
(XLSX)

**S2 Table. The configuration of each triangle used in experiment 2.**
(XLSX)

**S1 Fig.**  Raw trials from experiment 1 (top 8) and experiment 2 (bottom 8).
(TIF)

**S2 Fig. Comparing vision and non-vision trials.** (A) Combined distance walked during guided sides during vision on and vision off trial, showing now differences ($t(21) = 1.09$, $p = 0.288$, Cohen's $d = 0.336$ and $BF_{01} > 3$) (B) Angle error from experiment 1, showing a small but significant difference between vision on and off condition ($t(21) = 2.46$, $p < 0.022$, Cohen's $d = 0.248$ and $BF_{10} = 2.54$) (C) Distance error from experiment 1, showing a significant difference between vision on and off condition ($t(21) = 2.71$, $p < 0.013$, Cohen's $d = 0.232$ and $BF_{10} = 3.94$). (D) Angle error from experiment 1, ANOVA significant for triangle type $F(6,21) = 2.9$, $p < 0.01$, $\eta^2 = 0.058$ $BF_{10} = 1.72$. and Vision $F(1, 21) = 4.9$, $p < 0.026$, $\eta^2 = 0.016$ $BF_{10} = 1.16$, but not for the interaction between Type and Vision $F(6, 21) = 1.454$, $p = 0.194$, $\eta^2 = 0.029$ $BF_{10} = 0.432$. (E) Distance error from experiment 1, ANOVA significant for triangle type $F(6, 21) = 5.7$, $p < 0.1.33e-5$, $\eta^2 = 0.109$ $BF_{10} > 10$ and r Vision $F(1, 21) = 8.2$, $p < 0.004$, $\eta^2 = 0.026$ $BF_{10} > 4$, but not for the interaction between Type and Vision $F(6, 21) = 0.199$, $p = 0.976$, $\eta^2 = 0.004$ $BF_{10} > 10$.
(TIF)

**S3 Fig. Comparing right and left-handed trials.** (A) Angle error from experiment 1 which showed no difference between left and right-handed triangles ($t(21) = 0.7$, $p = 0.485$, Cohen's $d = 0.118$ and $BF_{01} > 3$). (B) Distance error from experiment 1, which showed no difference between left and right-handed triangle ($t(21) = 1.136$, $p = 0.268$, Cohen's $d = 0.103$ and $BF_{01} = 2.53$). (C) Angle error from experiment 2, again showing no difference between left and right-handed triangle ($t(16) = 1.51$, $p = 0.151$, Cohen's $d = 0.245$ and $BF_{01} = 1.55$). (D) Distance error from experiment 2, which showed no difference between left and right-handed triangle

($t(16) = 0.724$, $p = 0.4797$, Cohen's d = 0.176 and $BF_{01} = 3.188$).
(TIF)

**S4 Fig. Angle and distance accuracy.** Raster plot (A) showing the percentage of responses with less than 15% angle error (ranging from -27˚ to 27˚) for triangle type (x-axis) and participants (y-axis). Participants were 281.39% more likely to have <15% angle error in their unguided side than <15% total error (angle and distance). (B) percentage of responses with less than 15% distance error (8.5m to 11.5m). Participants are 208.14% more likely to have <15% distance error in their unguided side than <15% total error (angle and distance). (C) percentage of responses with less than 15% angle error (ranging from -27˚ to 27˚) and 10% distance error (8.5m to 11.5m). In (C) we can see that all of participant AT03's responses for triangle 45-90-45 are less than 15% error for both angle and distance error. And 80% for equilateral triangle (60-60-60) for participant AT11.
(TIF)

**S5 Fig. Position Error.** Mean position error (total distance from the participant's final position and the origin. No main effect of triangle type (left) 1-way ANOVA $F(6,21) = 1.34$, $p<0.24$, $\eta2 = 0.06$ $BF_{01} = 6.76$.
(TIF)

**S6 Fig. Best bit model between Model 1, Model 2, and Encoding-Error Model.** Comparing model fitting of the individual participant's data. A) Shows best model fit (highest loglikelihood values) for each subject in experiment 1. B&C) Lowest AIC and BIC values across the 3 models for each subject in experiment 2. D) Shows best model fit (highest loglikelihood values) for each subject in experiment 2. E&F) Lowest AIC and BIC values across the 3 models for each subject in experiment 2
(TIF)

**S7 Fig. Model recovery between Model 1, Model 2, and Encoding-Error Model.** Model recovery confusion matrices. In column and row, 1 = Model 1, 2 = Model 2, and 3 = the Encoding-Error Model. Probability ranges from 0 to 1. (A & B) Best AIC and BIC for Experiment 1 respectively. The higher value in the diagonal shows better model recovery from this experiment. The Encoding-Error does not fit its own simulated data well. (C)The Exceedance Probability for Experiment 1. (D & E) show best AIC and BIC for Experiment 2 respectively. Again, we see Encoding-Error does not fit its simulated data well. (C)The Exceedance Probability for Experiment 2.
(TIF)

**S8 Fig. Best bit model between Model 2 and Encoding-Error Model.** Comparing model fitting of the individual participant's data. A) Best model fit (highest loglikelihood values) for each participant in experiment 2. B&C) Lowest AIC and BIC values across the 3 models for each participant in experiment 2. D)The exceedance probability of each model for experiment 2.
(TIF)

**S9 Fig. Model recovery between Model 2 and Encoding-Error Model.** Model recovery confusion matrices. In rows and columns, 1 = Model 1 and 2 = the Encoding-Error Model. Probability ranges from 0 to 1. (A & B) Show best AIC and BIC for Experiment 2 respectively. We see the Encoding-Error does not fit its own simulated data well. (C)The Exceedance Probability for Experiment 2.
(TIF)

**S10 Fig. Vector addition model Encoding error model.** Recreation Fig 1 in Fujita et al. 1993 (modified to employ the names of variable used in our study), as shown with the black solid line which represents the participant's walked trajectories. The dashed blue lines are the internal encoded representation as predicted by the Encoding-Error Model (these are represented with subscript r). The dashed red line is a rough overlay for the paths predicted by the vector addition model. As shown below, the Encoding-Error model creates sides $A_r$ and $B_r$ and angle $\angle ab_r$. According to Euclidian properties, this results in side $C_r = C$ and angle $\angle bc_r = \angle bc$. The vector addition model (in red), instead, adjusts A and B accordingly to fit participant's response C. Thus, the angle $\angle ab$ and $\angle bc$ remain relatively the same in the vector addition model. The critical difference here is that the vector addition models assume a different suboptimal encoding of guided distance for sides A and B. In contrast, the Encoding-Error model assumes the same suboptimal encoding of the guided sides A and B and separate suboptimal encoding of $\angle ab$ such that it preserves Euclid's postulates. *Note that this does not include fitted noise which would slightly change the direction of all three sides. This would in fact make it non-Euclidean such that the total sum of all internal angles does not equal 180.
(TIF)

**S1 Text. Fitting Encoding-Error Model.**
(PDF)

## Acknowledgments

The Authors are gratful to E. Erlenbach for helping during data collection.

## Author Contributions

**Conceptualization:** Sevan K. Harootonian, Arne D. Ekstrom.

**Data curation:** Sevan K. Harootonian, Eli M. Ziskin.

**Formal analysis:** Sevan K. Harootonian.

**Funding acquisition:** Arne D. Ekstrom.

**Investigation:** Sevan K. Harootonian.

**Methodology:** Sevan K. Harootonian, Robert C. Wilson, Arne D. Ekstrom.

**Software:** Sevan K. Harootonian, Robert C. Wilson, Lukáš Hejtmánek.

**Supervision:** Robert C. Wilson, Arne D. Ekstrom.

**Writing – original draft:** Sevan K. Harootonian, Robert C. Wilson, Arne D. Ekstrom.

**Writing – review & editing:** Sevan K. Harootonian, Robert C. Wilson, Arne D. Ekstrom.

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
