## [Decision Letter · Decision Letter 0]

20 Nov 2019

Dear Dr Harootonian,

Thank you very much for submitting your manuscript 'Path integration in large-scale space and with novel geometries: Comparing Vector Addition and Encoding-Error Models' for review by PLOS Computational Biology. Your manuscript has been fully evaluated by the PLOS Computational Biology editorial team and in this case also by independent peer reviewers. The reviewers appreciated the attention to an important problem, but raised some substantial concerns about the manuscript as it currently stands. While your manuscript cannot be accepted in its present form, we are willing to consider a revised version in which the issues raised by the reviewers have been adequately addressed. We cannot, of course, promise publication at that time.

Sincerely,

Francesco P. Battaglia

Associate Editor

PLOS Computational Biology

Samuel Gershman

Deputy Editor

PLOS Computational Biology

[LINK]

Reviewer's Responses to Questions

**Comments to the Authors:**

Reviewer #1: ***I've also attached the review because it looks like some of the special characters didn't come through here***

This manuscript tested whether the shape and scale of the path during path integration leads to systematic errors. Using a novel omnidirectional treadmill technique and a triangle completion task, the authors manipulated the triangle shape while keeping the homing distance constant (Exp 1) and manipulated triangle size while keeping the shape constant (Exp 2). In Exp 1, participants overshot the turn angle and undershot the distance. Although the authors found significant differences among the range of triangle shapes, they did not find a systematic effect. In Exp 2, the turn angles were all similar, but participants underestimated distance, related to the scale of the triangle. They also made several model comparisons, determining that a vector addition model performed better than a previous model, the Encoding Error model. Overall, the vector addition model performed better, with a variant that included history of previous trials being the best fit for Exp 2. The authors conclude that vector-based models are a simple way to explain human path integration processes.

There is much to like in this manuscript, and overall it makes a positive contribution to the literature on path integration. It is nice to see the researchers using the omnidirectional treadmill in this novel way to answer questions that are intractable for standard setups. The experimental setups are elegant, and the detailed model comparison is important.

I have two main criticisms as well as some more medium/minor comments that could improve the manuscript.

1. More clarity is needed in the introduction to make the distinction between the vector addition and Encoding Error models. When the two models are first introduced (page 5), they seem like small variants of each other, but later (page 8) on the authors are setting up a contrast between the two. Yet the theoretical difference has not been fully made clear.

In addition, in the introduction it seems like the authors are confounding homing vector models and vector addition models, which are somewhat different. A homing vector (also called a continuous strategy), as it has been defined in the literature (e.g. Fujita et al, 1990; Wiener et al., 2010), is a continuous updating of the animal’s location relative to home, with a single vector representing the relationship to home. This model is history-free in that it does not (in theory) track anything about the path itself, just the return vector. Thus, a vector addition model is not quite the same thing as a homing vector model, because it requires memory of the vectors traveled during triangle completion.

In contrast, the Encoding Error model – and, from the description, the vector addition model – is a configural model, in which the shape of the outbound path is remembered. Most of the models described in the introduction are also configural models, not homing vector models. So while I think it is completely a good and valid question to compare vector addition and Encoding Error models, the distinction of these two within the family of configural models needs to be made clear, as well as the distinction with the family of homing vector models. This issue pops up again in the discussion on page 26 – again, more clarity on the distinction would help.

2. The results of Exp 1 are pretty clear and the design is easy to follow, but I disagree with the interpretation. The authors found significant differences by triangle type in the angular error and distance error. The conclusion that shape does not contribute to errors just doesn’t follow from those results. It does seem like shape might not contribute to systematic errors – or at least, not systematic errors that the authors can determine. There likely are systematic errors but they are just not obvious. But shape clearly is important and makes a difference to the errors in triangle completion. The authors seem to acknowledge this themselves in line 347, when saying that the consistency of the configuration was important for Exp 2.

Medium comments:

1. The section titles in the results seemed a bit too “conclusion-y” for a results section and seem better suited to the discussion (especially given the question about the conclusions in point 2 above). More descriptive section headers about the contents of the outcome measures would be appreciated.

2. For much of the results (as well as the figures), there is no mention of the statistical test used or what the design of the contrast is. It is also not listed in the methods. It seems like in some places there are portions of a 2-way ANOVA scattered around, but it was difficult to tell. Are these one-way tests against 0? Between triangle types? It was difficult to follow.

3. Throughout the results and methods, it was not entirely clear what the visual information was. Since the participants were wearing an HMD, it seemed at first like they would be seeing something (a ground plane?), but then when the comparison against the vision condition came up, I wasn’t so sure. Was it primarily a blacked out screen for most of the trials? More details of the vision condition (maybe a figure of what people saw) would be helpful – was it just the monolith or was there anything else in the environment?

4. It was not clear whether the number of trials (28 in Exp 1, 30 in Exp 2) was per triangle type or altogether. From the description of the trials in Exp 2 it looks like it might be altogether. That does not leave a lot of repeats for each triangle type (e.g. 4 in Exp 1), which could be tricky and underpowered to do the modeling. A few of the triangles in Exp 2 only had 2 trials, which is really difficult to be able to draw firm conclusions on – the authors note this themselves on page 20. Were the triangle types randomized or in a set order during the experiment? This is particularly important when it comes to modeling the history of previous trials.

5. Based on the discussion on page 12 and elsewhere, the authors may want to consider a measure that combines distance and angle. One possibility is position error, just the straight-line distance the person was away from the target.

6. I had a bit of difficulty following why the beta in line 392 was significant, but the larger beta from Exp 1 was not. Other people will probably wonder as well, so perhaps a bit of explanation about how the stats pan out here would be helpful.

7. The discussion on page 21 about the underestimation of the unguided leg with distance could be related to the execution of the homeward trajectory (assumption 4 of the Encoding Error model). A few people have looked at this idea, such as (Wan, Wang, & Crowell, 2013, Spatial Cog & Comp; Chrastil & Warren, 2017, Exp Brain Res).

8. For the claim on the last line of the main paragraph on page 22 (~line 471) seems like it needs more evidence to support it, especially given the distinction between homing vector and vector addition.

9. The discussion about the difference between circuitous paths on the treadmill vs. the Souman study (page 24) is lacking a mention of a clear difference between the two: the treadmill included the feedback from the controllers to keep people going in a straight line. It is difficult to accept that the interface made no difference when the controller was providing information.

10. Limitations of the vector model: It wasn’t clear what the authors meant by “directions”, and perhaps this exposes where more clarity is needed about the vector model. Does the vector addition include the direction of the vector or just distance? If they do not have a direction, how are they a vector? Please clarify this aspect of the model.

11. A little more clarity in the modeling section of the methods would help, just to make sure it’s clear what is in model 1 and what is in model 2. My reading is that model 1 is a weighted sum of the vectors, and model 2 is a weighted sum including noise whose variance is related to distance walked (on the outbound path?) and including a weighing of previous trials.

12. Figure 2: Explain what the statistical tests are and what they show (e.g. ANOVA examining differences between triangle types, and a significant effect of triangle type was found). Figure 3 could use more information in the caption. Figures 4-6: It might be clearer to arrange by Experiment rather than by model, so that it becomes 2 figures (one for Exp 1 and one for Exp 2). It is easier to compare the three models within an Experiment. Perhaps also show the actual data in that combined figure. In the supplement, S6 seems like it might contract the findings in the results. It is also a bit confusing about the difference between figures S6 and S8, as well as S9. Why is model 1 removed here?

13. Appendix A needs some additional clarity. Define �1, �3, and �4. Presumably these are the parameters you are fitting, but make it explicit. Explain why the same beta number (e.g. �3) appear in multiple equations. Explain more clearly what Level 1 and Level 2 are modeling, it currently doesn’t quite line up to be clear. What values are taken from the experimental design (such as the length of the outbound legs) and what values are taken from the data?

Minor comments:

1. Line 150, “pathway” should be “pathways”

2. Line 382, add that the beta values were significantly less than 1. (Was this a 1-sample t-test against 1?)

3. Line 562, “…plausible then…” should be “…plausible than…”

4. Line 652, how did the controller provide feedback? Vibrate on the side to turn towards or turn away from?

5. I suggest moving the remainder of the experimental methods before the modeling, because it seems to kind of break up the flow of the analysis.

6. In the modeling portion, there are a few places where the notation was not clear. When first introduced, xtc does not explain what the t superscript is – it comes out eventually in a few pages but could use at least a mention early on that it is the current trial. Later on, � looks a lot like x (it took me a while to realize they were different), so you might want to consider a different symbol. Similarly, � and � were used in the description of the triangles (e.g. Figure 1), but were also used in the modeling portion to refer to something different. This could create confusion.

Reviewer #2: Uploaded as attachment

**Have all data underlying the figures and results presented in the manuscript been provided?**

Reviewer #1: None

Reviewer #2: Yes

PLOS authors have the option to publish the peer review history of their article (what does this mean?). If published, this will include your full peer review and any attached files.

Reviewer #1: No

Reviewer #2: No

---

## [Decision Letter · Decision Letter 1]

10 Mar 2020

Dear Research Specialist Harootonian,

Thank you very much for submitting your manuscript "Path Integration in Large-Scale Space and with Novel Geometries: Comparing Vector Addition and Encoding-Error Models" for consideration at PLOS Computational Biology. As with all papers reviewed by the journal, your manuscript was reviewed by members of the editorial board and by several independent reviewers. The reviewers appreciated the attention to an important topic. Based on the reviews, we are likely to accept this manuscript for publication, providing that you perform the edits suggested by one of the reviewers.

Sincerely,

Francesco P. Battaglia

Associate Editor

PLOS Computational Biology

Samuel Gershman

Deputy Editor

PLOS Computational Biology

[LINK]

Reviewer's Responses to Questions

**Comments to the Authors:**

Reviewer #1: The authors have done an excellent job responding to my comments, and I recommend the revised manuscript for publication.

Reviewer #2: Comments uploaded as attachment

**Have all data underlying the figures and results presented in the manuscript been provided?**

Reviewer #1: None

Reviewer #2: Yes

PLOS authors have the option to publish the peer review history of their article (what does this mean?). If published, this will include your full peer review and any attached files.

Reviewer #1: No

Reviewer #2: No
---

## [Editor Report · Decision Letter 2]

24 Mar 2020

Dear Research Specialist Harootonian,

We are pleased to inform you that your manuscript 'Path Integration in Large-Scale Space and with Novel Geometries: Comparing Vector Addition and Encoding-Error Models' has been provisionally accepted for publication in PLOS Computational Biology.

Best regards,

Francesco P. Battaglia

Associate Editor

PLOS Computational Biology

Samuel Gershman

Deputy Editor

PLOS Computational Biology

---

## [Editor Report · Acceptance letter]

15 Apr 2020

PCOMPBIOL-D-19-01761R2 

Path Integration in Large-Scale Space and with Novel Geometries: Comparing Vector Addition and Encoding-Error Models

Dear Dr Harootonian,

I am pleased to inform you that your manuscript has been formally accepted for publication in PLOS Computational Biology. Your manuscript is now with our production department and you will be notified of the publication date in due course.

With kind regards,

Laura Mallard
